# The Spectrum of Small Heat Shock Protein B8 (*HSPB8*)-Associated Neuromuscular Disorders

**DOI:** 10.3390/ijms26072905

**Published:** 2025-03-23

**Authors:** Hebatallah R. Rashed, Samir R. Nath, Margherita Milone

**Affiliations:** 1Department of Neurology, Mayo Clinic, Rochester, MN 55905, USA; rashed.hebatallah@mayo.edu (H.R.R.); nath.samir@mayo.edu (S.R.N.); 2Department of Neurology, Ain Shams University, Cairo 11588, Egypt

**Keywords:** CASA, CTM2, CMT2L, dHMN, HMN, HSPB8, myofibrillar myopathy, myopathy, rimmed vacuoles

## Abstract

The heat shock protein B8 (HSPB8) is one of the small heat shock proteins (sHSP or HSPB) and is a ubiquitous protein in various organisms, including humans. It is highly expressed in skeletal muscle, heart, and neurons. It plays a crucial role in identifying misfolding proteins and participating in chaperone-assisted selective autophagy (CASA) for the removal of misfolded and damaged, potentially cytotoxic proteins. Mutations in *HSPB8* can cause distal hereditary motor neuropathy (dHMN), Charcot–Marie–Tooth (CMT) disease type 2L, or myopathy. The disease can manifest from childhood to mid-adulthood. Most missense mutations in the N-terminal and α-crystallin domains of HSPB8 lead to dHMN or CMT2L. Frameshift mutations in the C-terminal domain (CTD), resulting in elongation of the HSPB8 C-terminal, cause myopathy with myofibrillar pathology and rimmed vacuoles. Myopathy and motor neuropathy can coexist. HSPB8 frameshift mutations in the CTD result in HSPB8 mutant aggregation, which weakens the CASA ability to direct misfolded proteins to autophagic degradation. Cellular and animal models indicate that *HSPB8* mutations drive pathogenesis through a toxic gain-of-function mechanism. Currently, no cure is available for *HSPB8*-associated neuromuscular disorders, but numerous therapeutic strategies are under investigation spanning from small molecules to RNA interference to exogenous HSPB8 delivery.

## 1. Introduction

The heat shock protein B8 (HSPB8) is one of the ten small heat shock proteins (sHSP or HSPB) present in mammalians, including humans, named HSPB1 through 10. They are a group of highly conserved chaperones with low molecular weight [1,2]. Like other HSPBs, HSPB8 is an ATP-independent holdases that prevents aggregation of misfolded proteins and participates in the refolding process while interacting with ATP-dependent foldases, such as heat shock protein 70 (HSP70 or HSPA). Therefore, HSPB8 plays an essential role in protein quality control. It also participates in chaperone-assisted autophagy, a crucial process for the removal of misfolded and damaged potentially cytotoxic proteins [3]. HSPB8 is a ubiquitous protein, highly expressed in skeletal muscle, heart, neurons, glial cells, and, at a lower level, in other tissues, such as lung and kidney. Therefore, it is not surprising that *HSPB8* mutations cause neuromuscular disorders. Following the detection of the first *HSPB8* mutations in distal hereditary motor neuropathy (dHMN), discovered simultaneously with *HSPB1* mutations as cause on another dHMN [4,5], mutations in *HSPB8* have also been linked to Charcot–Marie–Tooth (CMT) disease and muscle disease. Herein, we describe the spectrum of neuromuscular disorders of variable onset and phenotype stemming from mutations in *HSPB8*.

## 2. Distal Hereditary Motor Neuropathy (dHMN)

### 2.1. Clinical Features

dHMN and distal spinal muscular atrophy (distal SMA) are synonyms for a spectrum of clinically and genetically heterogeneous disorders characterized by the degeneration of the motor axons, leading to distal muscle weakness and atrophy in the absence of sensory involvement [6,7,8]. Age of onset is variable and sometimes additional neurological features are present. More than 30 genetically distinct forms of dHMN have been recognized, including *HSPB8*-dHMN, which is typically due to autosomal dominant mutations, though most patients remain without a genetic diagnosis [8,9,10].

*HSPB8* was first identified as causative of autosomal dominant dHMN in 2004, when two different autosomal dominant mutations (c.423G->C, p.Lys141Asn and c.421A->G, p.Lys141Glu) in this gene were detected in a Czech family and a Belgian family after the disease had been mapped to a 5-Mb candidate region at 12q24.3 in the Belgian family [4,7,11,12]. The age of onset was between mid-teens and mid-20s. The weakness was lower limb predominant and initially affected the big toe extensors to later involve tibial and peroneal muscles. Patients developed distal lower limb paralysis within 10 years from onset. Weakness and atrophy progressed to involve thigh muscles and distal upper limb muscles. Some patients experienced muscle cramps and fasciculations in muscles undergoing atrophy. Sensory abnormalities were typically absent, though a subset of older patients showed decreased vibratory sense. Tendon reflexes were reduced or absent in the lower limbs. More recently, four additional *HSPB8* mutations were identified in 7 of 510 patients with dHMN [13]. The age of onset spanned from 8 to 60 years. All patients had slowly progressive distal lower limb weakness, and two also had distal upper limb weakness. No patient had proximal weakness. Two patients reported feet paresthesia. Most patients had pes cavus. No patient showed evidence of central nervous system involvement. Table 1 summarizes the clinical findings of HSPB8-associated neuromuscular disorders.

Analysis of the genetic etiology of 112 European index patients with dHMN only detected *HSPB8* pathogenic variants in 3 patients [14]. A study on hereditary motor neuropathies in North England estimated that dHMN has a prevalence of 2.14 affected individuals per 100,000 people in that geographic area, but only 1 of the 64 described patients had *HSPB8*-dHMN, suggesting that this is one of the least frequent dHMNs [15]. In agreement with such finding, an observational study on dHMN, conducted in two tertiary neuromuscular referral centers in Spain, identified no patients with *HSPB8*-dHMN among 163 affected individuals from 108 different families [8]. A single patient of this cohort carried a variant of unknown significance (c.65G>A, p.Arg22Gln) in *HSPB8*.

### 2.2. Serology Studies

The creatinine kinase (CK) level was investigated in one patient who showed mild hyperCKemia (530 U/L; N < 200) [13]. No myopathic electromyographic findings were reported in this patient.

### 2.3. Electrophysiology Studies

Electrophysiological studies typically show predominantly distal and symmetric chronic neurogenic findings with minimal or no fibrillation potentials, suggestive of a motor axonal neuropathy [12,13,16]. A mild axonal sensory involvement can occur in some patients [13], demonstrating an overlap between dHMN and CMT2L.

### 2.4. Radiology Studies

Brain and spinal cord MRI examination performed in very few patients with *HSPB8*-dHMN was normal [13].

### 2.5. Pathology Findings

A nerve biopsy of the superficial peroneal sensory nerve in one patient with dHMN showed no remarkable abnormalities under light and electron microscopy; exceedingly rare onion bulbs were detected after extensive search [11,16]. This patient’s peroneus brevis biopsy demonstrated fatty replacement of muscle and a few atrophic fibers.

## 3. Charcot–Marie–Tooth Disease Type 2L (CMT2L)

### 3.1. Clinical Features

CMT or hereditary motor and sensory neuropathy (HSMN) is the most common inherited peripheral nerve disorder [17]. CMT is traditionally classified based on nerve conduction velocity in demyelinating (CMT1), axonal (CMT2), and dominant intermediate. CMT2 is further categorized into various subtypes with respect to the genotype [17].

In 2004, one of two *HSPB8* mutations (c.423G->T. p.Lys141Asn) previously discovered in *HSPB8*-dHMN was identified as causative of CMT2L in a large Chinese family previously mapped to a 6.8-cM candidate region at 12q24 [18,19]. The age of onset of the neuropathy varied from mid-teens to early 30s and presented with a slowly progressive lower limb predominant distal muscle weakness and atrophy. Two patients also had proximal weakness. Patients had associated sensory loss and hypo- or areflexia. Most patients demonstrated pes cavus, and a few had scoliosis (Table 1). In 2013, a novel *HSPB8* mutations was discovered in a 27-year-old Korean patient with CMT2 who presented with a similar phenotype [20]. These patients did not have painless ulcerations or injuries.

*HSPB8*-CMT2L remains very rare. No patients with *HSPB8* mutations were identified in a cohort of 61 patients from 18 families with CMT2 [21]. In general, *HSPB8* mutations account for 1% of patients with CMT2/dHMN [13].

### 3.2. Electrophysiology Studies

Electrophysiological studies show normal nerve conduction velocities but reduced or absent compound muscle action potentials (CMAPs) and sensory nerve action potentials (SNAPs). Electromyography reveals neurogenic motor unit potentials, fibrillation potentials, and positive sharp waves. The electrophysiological findings are consistent with a chronic axonal sensorimotor polyneuropathy [18,19,20].

### 3.3. Radiology Studies

Muscle MRI in one patient revealed signal abnormality in the lower limb muscles. T1-weighted images demonstrated severe muscle atrophy and fatty replacement in the leg muscles, compared to the thigh muscles. There was, however, also fatty involvement of the anterior and posterior thigh muscles, especially of the vastus muscle group. Gracilis, rectus femoris, and medial compartment muscles were relatively spared [20].

### 3.4. Nerve Pathology Findings

Nerve biopsies show features of axonal degeneration, loss of large, myelinated fibers, and occasional clusters of thinly myelinated axons [18,19,20]. Electron microscopy studies reveal vacuolization, focal granular material aggregation, and organelles loss within the axons of myelinated fibers. Schwann cells with features of degeneration and vacuolated unmyelinated axons can also be observed.

## 4. *HSPB8*-Myopathy

### 4.1. Clinical Features

More recently, mutations in *HSPB8* were recognized as the cause of an autosomal dominant myopathy with myofibrillar pathology and rimmed vacuoles. *HSPB8* myopathy was first described in 2016 in two families presenting with a distal neuromyopathy phenotype [22]. Since then, only a few additional patients with *HSPB8*-myopathy have been reported [23,24,25,26,27,28,29]. Since the initial submission of this review, *HSPB8*-myopathy has been referenced in the Online Mendelian Inheritance in Man (OMIM) as myofibrillar myopathy type 13 (MFM13) with rimmed vacuoles.

The myopathy usually manifests from adolescence to mid-adulthood, but childhood onset can occur with the youngest reported patient developing weakness at age 6 (Table 1) [29]. Lower limbs are often more affected than upper limbs muscles. At onset, there is typically distal lower limb weakness with involvement of the ankle dorsiflexors, later extending to proximal lower and upper limb muscles as well as axial muscle. Weakness can also manifest in proximal lower limb muscles (Figure 1). The gastrocnemius is relatively spared. Axial weakness resulting in camptocormia can also be the initial manifestation of the myopathy, followed by limb girdle weakness. The weakness, which is accompanied by development of muscle atrophy, can be asymmetric. The abdominal muscles weakness can lead to detection of Beevor sign. Facial weakness has not been observed. Tendon reflexes may be normal, diminished, or absent. Patients may show skeletal abnormalities, such as lumbar lordosis, scoliosis, or pes cavus with hammertoes.

Muscle cramps and fasciculations can occur early in the disease course, especially in patients with associated neurogenic changes. Coexisting motor neuropathy has been observed in several patients [22,23,25].

Patients may develop respiratory dysfunction ranging from asymptomatic to severe respiratory failure. Some patients may require non-invasive ventilatory support or even mechanical ventilation [25,27,28]. In affected individuals of a family, respiratory failure requiring mechanical ventilation, manifested after a disease duration spanning from approximatively a decade to 40 years [27].

Cardiac symptoms are uncommon, and cardiomyopathy has been reported in a single family [28]. A patient had asymptomatic right bundle branch block on electrocardiogram [25].

HSPB8 myopathy is very rare. Its prevalence is unknown as very few families and sporadic cases have been reported so far.

### 4.2. Serology Studies

CK levels are normal or elevated up to approximately 2000 IU/L [22,23,25,26,28].

### 4.3. Electrophysiology Studies

Needle electromyography shows myopathic changes with rapid recruitment of short-duration motor unit potentials with or without fibrillation potentials [23,25,26,28]. Intermixed long-duration motor unit potentials are often detected in proximal and more frequently in distal muscles [22,24,26]. Nerve conduction studies are either normal or show an associated motor neuropathy with axonal features [22,23,24,25,26,28].

### 4.4. Radiology Studies

Muscle MRI demonstrates muscle signal changes early in the disease stage later progressing to fatty replacement [22,24,25,26]. Fatty changes of paraspinal muscles [23,25,26], and symmetric or asymmetric multifocal fatty degenerative changes of the lower limb muscles have been described [22,23,24,25,28]. In the thighs, the most affected muscles include the vastus muscles, especially vastus medialis and intermedius, adductor magnus, hamstrings, sartorius, and gracilis [22,23,24,25,28]. Some patients have shown a selective pattern of fatty degeneration of the adductor magnus and adductor longus and less involvement of adductor brevis, sartorius, gracilis, and hamstrings [24]. In the leg, tibialis anterior, extensor digitorum longus, gastrocnemius, and peroneus muscles are affected [22,23,24]. Some patients show predominant fatty replacement of the anterior and lateral compartment muscles and relatively spared soleus and gastrocnemius [22,25,28]. However, fatty degeneration and volume loss of tibialis posterior, both heads of gastrocnemius, and peroneus longus with sparing of other muscles can also occur [24].

### 4.5. Muscle Pathology Findings

Muscle pathology often shows dystrophic changes with features of myofibrillar pathology and rimmed vacuoles of various degree (Figure 2), depending on the underlying genetic defect [22,23,24,25,26,28,29]. Structural abnormalities can be patchy and the distribution of rimmed vacuoles or myofibrillar pathology may vary even within the same muscle biopsy. Myopathological findings include muscle fiber size variation, fiber atrophy, nuclei internalization, fiber splitting, cytoplasmic bodies, cytoplasmic inclusions, rimmed and non-rimmed vacuoles, sarcoplasmic masses, and increased endomysial fibrous connective tissue. Regenerating and necrotic fibers are either absent or rare. Core-like structures and moth-eaten fibers can be observed on oxidative stains.

By immunohistochemical or immunofluorescent studies, muscle biopsies may show sarcoplasmic protein aggregates immunoreactive for sarcomeric proteins, such as myotilin, desmin, αB-crystallin, but also dystrophin and even punctate T-cell intracellular antigen 1 (TIA1) positivity (Figure 2). Ectopic TAR DNA-binding protein 43 (TDP-43) positivity, SMI-31, sequestosome 1 (SQSTM1)/p62, and microtubule-associated protein 1 light chain 3B (LC3B) positive inclusions can be detected. In some patients, muscle biopsy displayed co-localization of HSPB8, DNAJB6, myotilin, BAG3, and TDP-43 within the myofibrillar aggregates, while rimmed vacuoles were p62-positive, and atrophic fibers were HSPB8 overexpressing [22].

Electron microscopy studies show focal disruption and loss of the myofibrils, accumulation of granulofilamentous or amorphous material, and Z lines streaming [22,23,24,29].

Fibroblasts from a patient with *HSPB8*-myopathy showed reduced expression of HSPB8 and increased expression of both the autophagosomal marker LC3B and autophagy receptor p62, compared to control fibroblasts [25].

## 5. Gene and Protein Structure and Function

*HSPB8* (also known as *HSP22*) is located on chromosome 12q24.23 and consists of three exons. The encoded HSPB8 is composed of 196 amino acids. HSPB8 is one of the ten human small heat shock proteins, which are ATP-independent molecular chaperones. They are structurally characterized by the α-crystallin domain (ACD), a highly conserved sequence of about 85 amino acids, which is flanked by the more variable hydrophobic N-terminal domain (NTD) and short C-terminal domains (CTD) (Figure 3) [1]. The ACD, which in HSPB8 consists of beta strands, is responsible for the protein’s core chaperone activity [30,31,32]. The NTD contains the conserved RLFDQxFG motif, which is crucial for the interaction with other heat shock proteins [33]. The CTD interacts with the exposed hydrophobic regions of substrates and inhibits their aggregation. Contrary to several other HSPBs that form oligomers through the I/V-X-I/V motif, HSPB8 lacks this motif and preferentially forms homodimers [34]. A hydrophobic pocket within the ACD of HSPB8 allows its binding to the two isoleucine-proline-valine (IPV) domains of the co-chaperone BAG3. Two HSP8 molecules bind one BAG molecule and form a chaperone complex [34,35,36].

As previously described, HSPB8 is an ATP-independent holdases that prevents aggregation of misfolded proteins. It also participates in the refolding process while interacting with ATP-dependent foldases, such as heat shock protein 70 (HSP70 or HSPA) [1,2,37]. HSPB8, like other HSPBs, has a crucial role in cellular protein quality control. Additional HSPB functions include response to cellular stress and processes favoring apoptosis, facilitation of substrates degradation, and maintenance of cytoskeleton structural integrity. HSPB8, together with BAG3, is a key component of the chaperone-assisted selective autophagy (CASA), a selective autophagic pathway for the disposal of misfolded and damaged proteins (Figure 4) [38,39]. CASA requires the formation of a heteromeric complex consisting of HSPB8, BAG3, the ATP-dependent chaperone HSPA (HSP70), and the E3 ubiquitin ligase STUB1, although other ubiquitin ligases may replace STUB1. BAG3 operates as scaffold for the interaction of the other CASA components; HSPB8 and HSPA recognize the misfolded and damaged proteins, which undergo HSPA-dependent refolding or STUB1-dependent ubiquitination. Ubiquitinated targets are then routed to the autophagosome lysosomal pathway for degradation through the involvement of the selective autophagy receptor SQSTM1/p62 [40]. Failure of this quality control machinery results in the accumulation of damaged and misfolded proteins in cytoplasmic aggregates. The CASA complex has been shown to have a crucial role in the degradation of damaged Z disc constituents in skeletal muscle and removal of mutated or cytotoxic protein aggregates in neurodegenerative diseases [41,42,43]. Regarding neuromuscular diseases, not only mutations in *HSPB8*, but also mutations in *BAG3* can cause myopathy with myofibrillar pathology (like that observed in *HSPB8* myopathy) or axonal neuropathy, further supporting the crucial role of the CASA complex in maintaining the functional integrity of the neuromuscular system, including sarcomer [44]. So far, no STUB1 or HSPA mutants have been associated with neuromuscular diseases, but mutations in *STUB1* have been recognized as cause of spinocerebellar ataxia [45].

HSPB8 is also involved in the cytosolic unfolded protein response (cUPR), a cellular response to proteotoxic stress and cytosolic protein aggregates driven by the heme-regulated kinase inhibitor [46].

HSPB8 has a role in stress granules dynamics and maintenance [38]. Stress granules are membrane-free ribonucleoprotein aggregates that form in response to proteotoxic stress when translation is inhibited. HSPB8 prevents irreversible aggregation of misfolded proteins in the stress granules and recruits BAG3 and HSPA to degrade the misfolded protein. HSPB8, in concert with BAG3 and HSPA, plays a role in stress granule surveillance. Considering this HSPB8 role, it is not surprising to also see small TIA1 positive inclusions in *HSPB8* myopathy [26].

Of interest, mutations in other HSPBs, in addition of *HSPB8*, cause neuromuscular diseases. For example, *HSPB1* mutations are a common cause of dHMN and CMT2 and can cause neuromyopathy and distal myopathy featured by rimmed vacuoles [5,47,48,49,50]. *HSPB3* mutations have been linked to dHMN and CMT2 [51,52]. *HSPB5* (alpha-B crystallin, *CRYAB*) mutations can cause myopathy with myofibrillar pathology or CMT2 [53,54].

## 6. Genetic Aspects and Pathomechanisms

The inheritance pattern of *HSPB8* mutations leading to neuromuscular disorders is autosomal dominant [4,11,12,13,16,22,23,25,27,28]. De novo mutations can occur [13,20,26].

The *HSPB8* mutations reported so far and the associated phenotypes are summarized in Figure 3. Most mutations occur in the ACD and CTD of the protein. A single mutation (p.Pro90Leu) causing dHMN has been described in the NTD [4,11,13]. Lysine141 in ACD is a hot spot and mutations affecting this residue can lead to dHMN, CMT2L, or neuromyopathy [4,13,19,20]. p.Lys141Asn causes either dHMN or CMT2L [4,11,13], underscoring the value of the rigorous clinical and electrophysiological characterization of patient’s phenotype for the interpretation of the genetic findings. Only two mutations have been associated with CMT2L and affect the Lys141 residue (p.Lys141Asn and p.Lys141Thr) [18,19,20]. Mutations causing myopathy mainly reside in the CTD and result in extension of the C-terminal of the protein [23,25,26,29]. Two of these mutations were associated with proximal weakness at disease onset, with one patient manifesting at age 19 (p.Thr194Serfs∗23) [26], and the other at age 6 (p.G192Afs*55) [29]. A third mutation (p.Gln170Glyfs*45) was linked with axial weakness at onset [23]. The remaining variants occurred in patients with distal onset of the weakness. The only mutation associated with myopathy, in the setting of neuromyopathy, and not located in CTD, is p.Lys141Glu within the ACD [4,22,24].

HSPB8 missense mutations involving Lys.141 were shown to reduce HSPB8 chaperone activity [3]. Comparison of the structural properties of the mutant HSPB8 (p.Lys141Glu), which cause dHMN or neuromyopathy, with the wild-type HSPB8 suggested that the mutation may result in destabilization of the HSPB8 structure and reduction of its chaperone function [3,55,56].

Regarding the *HSPB8* frameshift mutations, initial studies had suggested the possibility that the myopathy may be caused by HSPB8 haploinsufficiency by induction of nonsense-mediated mRNA decay or protein degradation [23]. More recently, elegant studies have demonstrated that HSPB8 frameshift mutants lead to a toxic gain-of-function [23,57]. They are insoluble and form cytoplasmic aggregates while continuing to retain their property to interact with the CASA components. This results in sequestration of the CASA components, as well as the autophagy receptors SQSTM1/p62, into the HSPB8 mutant aggregates, reduced CASA efficiency and proteostasis response failure. Additionally, HSPB8 frameshift mutants impair myoblast differentiation and fusion into myotubes, and compromise sarcomeric protein homeostasis.

## 7. Animal and In Vitro Models

The role of HSPB8 in promoting chaperone activity and chaperone mediated selective autophagy is well established across several experimental models. Functional analyses of HSPB8 mutations in vitro have showed a loss of chaperone activity and pro-degradative function of HSPB8 [3,55,58]. Carra et al. investigated the capacity of HSPB8 to prevent protein aggregation in the cells using the polyglutamine protein Htt43Q as a model. In most cellular models, HSPB8 blocked inclusion formation and maintained Htt43Q in a soluble state competent for rapid degradation [3]. More recently, Crippa et al. reported that HSPB8 suppresses the accumulation of aberrantly localized misfolded forms of TDP-43 and its 25 kDa C-terminal fragment (TDP-25), enhancing their clearance via autophagy [58].

Mutations of the HSPB8^K141^ residue cause multiple defects in HSPB8 function, including abnormal interactions of HSPB8 with other HSPB, including HSPB1, and impairment of chaperone activity and autophagosomes-lysosome delivery, leading to accumulation of misfolded proteins [59,60]. In primary rat motor neuron culture, expression of HSPB8 K141E as well as K141N resulted in both neurite degeneration and formation of spheroids within neurites [61]. Kim et al. demonstrated that HSPB8^K141E^ induces destabilization of the HSPB8 structure, which may diminish its chaperone-like activity with certain substrates [55]. More recently, Sisto et al. showed that mouse embryonic fibroblasts isolated from a HSPB8^K141N^ green fluorescent protein (GFP)-LC3 model have diminished autophagosome production when the mechanistic target of rapamycin kinase (MTOR) pathway is inhibited [56]. Notably, *HSPB8* knock-out mice show locomotor performance comparable to wild-type mice, along with normal sciatic nerve morphology and preserved myofiber and Z-line organization, suggesting that a toxic gain-of-function of mutant HSPB8, rather than loss-of-function, drives the neuromuscular phenotype [60]. However, muscle pathology showed accumulation of abnormal mitochondria, which is in line with the functional alterations in cardiac mitochondria previously reported by Qiu et al. [62].

Various mouse models have been generated to study the effects of *HSPB8* mutations in vivo [60,62,63,64]. Of note, the HSPB8^K141N^ knock-in (KI) mouse model recapitulates the human disease phenotype, with homozygous KI mice developing a motor neuronopathy with progressive axonal degeneration [60]. The sciatic nerve of these mice was characterized by low autophagy potential in pre-symptomatic mice and HSPB8 aggregates in symptomatic mice. Functionally, these mice display declining strength and locomotor performance from about 9 months of age onward [60]. Similar electrophysiological, neuropathological, and locomotor deficits were seen in a transgenic mouse model ubiquitously overexpressing human HSPB8^K141N^ [64]. Alongside neuropathy, homozygous HSPB8^K141N^ KI mice developed a myopathy phenotype independent of the motor neuron pathology [60]. In line with human patients with *HSPB8*-myopathy, this myopathy resembles human myofibrillar myopathy with Z-disc disorganization, rimmed vacuoles, protein aggregates containing HSPB8, αB-crystallin, and desmin, as well as accumulation of LC3 and SQSTM1/p62 suggestive of possible impairment in the autophagic flux [60]. Although heterozygous HSPB8^K141N^ KI mice appear physiologically normal in functional tests, ultrastructural analysis of nerve and muscle revealed similar pathological changes as those seen in homozygotes.

The *Drosophila Hsp67Bc* gene encodes a protein belonging to the HSPB family localized at the Z- and A-bands and neuromuscular junctions. It is the nearest functional ortholog of the human *HSPB8* [65,66]. Using this *Drosophila* model, the effects of muscle-specific overexpression of *Drosophila* Hsp67Bc hot-spot variants Hsp67Bc R126E and Hsp67Bc R126N (corresponding to HSPB8 p.K141E and p.K141N, respectively) yielded distinct phenotypic outcomes [66]. The p.R126E mutant flies exhibited impaired motility, which may result from myofibrillar disorganization, altered neuromuscular junctions (decreased number of synaptic boutons), mitochondrial disruption and depolarization. In contrast, the p.R126N mutation led to excessive aggregation of the mutant protein but less severe sarcomeric alterations without mitochondrial abnormalities. In functional assays, these flies showed functional performance like wild-type flies [66].

Taken together, studies of animal and cellular models indicate that *HSPB8* mutations drive pathogenesis through a toxic gain-of-function mechanism involving disrupted chaperone function and impaired autophagic clearance of misfolded proteins.

## 8. Therapeutic Strategies

Given its key role in preventing aggregation and favoring misfolded protein degradation, modulation, or induction, HSPB8 may represent a promising therapeutic approach in proteinopathies and protein-folding disease.

Enhancing autophagy to increase misfolded protein degradation is an area of active research in chaperonopathies, proteinopathies, and other neurodegenerative diseases [67,68]. Studies have been conducted to identify small molecules able to induce, stabilize, or inhibit HSPB8 to potentiate the clearance of mutated proteins in neurodegenerative disorders, such as amyotrophic lateral sclerosis, Parkinson’s disease, and Alzheimer’s disease. When neuronal cells expressing the *HSPB8* promotor upstream of a luciferase enzyme were used for high-throughput screening, doxorubicin and colchicine were identified as dose-dependent inducers of HSPB8 [36,58]. Trehalose, an inducer of autophagy, has also been shown to increase HSPB8 expression. This induction of autophagy is mediated through transcription factor EB (TFEB), though induction of HSPB8 by trehalose was TFEB independent in cellular models [69,70,71] Recently, piplartine was shown to enhance autophagy in cellular HSPB1 and HSPB8 mutant models [56]. Geranylgeranylacetone, also known as teprenone, broadly induces heat shock protein expression and increases HSPB8 expression exhibiting a protective effect on the vascular endothelium of mice.

Recently, RNA interference targeting HSPB8 has been investigated to ameliorate the neuronal and muscle phenotype in a mouse model of *HSPB8*-dHMN but failed to show improvement of function and neuropathological findings [72]. The study led to the conclusion that the RNA interference induced HSPB8 reduction may not be an ideal therapeutic option, unless one would introduce a higher viral load and early treatment.

Another promising therapeutic strategy is the delivery of exogenous HSPB8 into target cells and tissues using viral vectors and HSPB8-loaded extracellular vesicles. Oligodendrocyte derived vesicles loaded with HSPB8 were recently shown to reduce ubiquitinated proteins and ROS in microglia and primary mixed neural cultures [73].

The growing evidence supporting both pharmacological induction and exogenous delivery of HSPB8 underscores its potential as a therapeutic target in diverse proteinopathies and neurodegenerative disorders and remains an important future direction for the field.

## 9. Potential Limitations of Current Studies

Due to the rarity of *HSPB8*-associated neuromuscular disorders, it is possible that the full spectrum of clinical and pathological manifestations has not yet fully emerged. Additionally, being HSPB8 ubiquitously expressed, one cannot exclude that mutations in the encoding gene could affect other organs and tissues, resulting in additional diseases or contributing to the clinical phenotype associated with primary mutations in other genes. Currently available animal models and in vitro models do not express the patient’s whole genetic make-up. Gene-by-gene interactions, genetic modifiers, and gene–environment interaction can impact disease manifestation and its variability even within the same family regarding phenotype, age of disease onset, or severity. Although tremendous progress has been made to understand *HSPB8* disease mechanisms and creating disease models, current studies may have revealed only part of the underlying processes leading to disease. Further investigations are needed to unravel what causes muscle and nerve degeneration and muscle weakness. These are crucial steps for the development of specific therapies that could halt disease progression, reverse motor deficits, and prevent disease development. Similarly, results from studies investigating various therapeutic options must take into account the limitations of the current disease models, in addition to potential side effects and drug toxicity in humans. Additionally, one should also consider that the rarity of *HSPB8*-associated neuromuscular disorders and their clinical heterogeneity will limit the ability to assess clinical response to future treatment.

## 10. Conclusions

The spectrum of neurological disorders associated with *HSPB8* is in continuous expansion and can involve motor axons, peripheral nerves, and muscles, in isolation or in combination. The clinical phenotype of the myopathy is variable with distal, proximal, or axial weakness. Patients may develop respiratory muscle weakness and, therefore, require periodic monitoring of their respiratory status. Although cardiomyopathy has been reported only in a family, considering that HSPB8 is expressed in cardiac muscle and that cardiomyopathy occurs in a mouse model of *HSPB8* disease, screening for underlying cardiac involvement should be part of patient care. Apart from rare exceptions, missense mutations tend to be associated with dHMN or CMT2L while frameshift mutations in the C-terminal domain of HSPB8, leading to C-terminal elongation, result in myopathy with myofibrillar pathology and rimmed vacuoles. In such cases, the mutant HSPB8 aggregates, impairs CASA and downstream events leading to compromised proteostasis, and alters the sarcomere organization. Currently no treatment is available for *HSPB8*-related neuromuscular disorders to ameliorate patients’ motor deficit or halt disease progression, though multiple therapeutic strategies are under investigation.

## Figures and Tables

**Figure 1 ijms-26-02905-f001:**
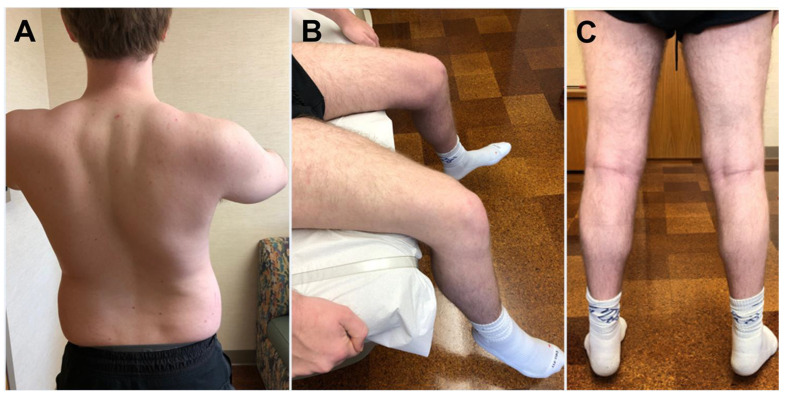
Patient with *HSPB8*-myopathy. Patient manifested with proximal lower limb muscle weakness that later extended to upper limb and axial muscles. (**A**) He had difficulty elevating the arms and scoliosis. (**B**) Quadriceps is atrophic and weak as suggested by his difficulty extending the knees. (**C**) Calf muscles, although atrophic, had preserved strength, and he was able to stand on toes. (This patient was previously reported: Nicolau et al. [24]).

**Figure 2 ijms-26-02905-f002:**
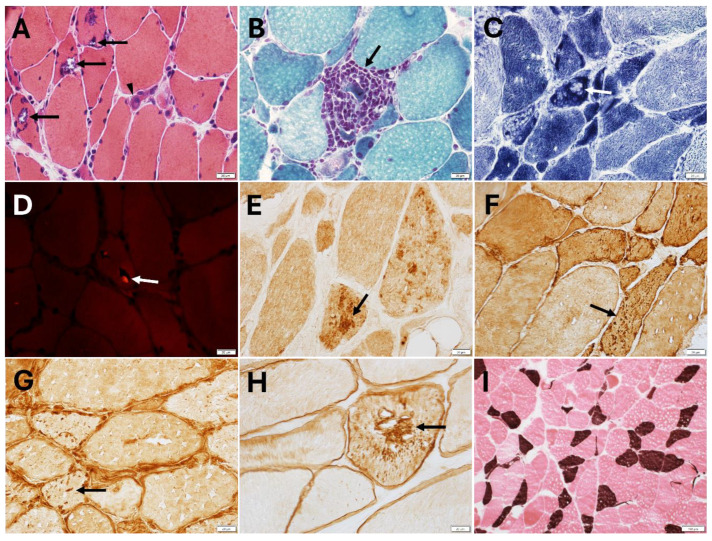
Quadriceps biopsy from patient with *HSBP8*-myopathy (p.Thr194Serfs∗23). (**A**) Many fibers harboring rimmed vacuoles (arrows) unevenly distributed across the muscle section and a regenerating fiber (arrowhead) (hematoxylin-eosin). (**B**) A necrotic fiber invaded by macrophages (arrow; trichrome). (**C**) In a region of the specimen, a few fibers showing a focal loss (arrow) or focal increase of nicotinamide adenine dinucleotide tetrazolium reductase (NADH) enzyme reactivity. (**D**) One (arrow) of the rare fibers containing two small congophilic inclusions (bright red; Congo red). A few fibers with myotilin aggregates (**E**), punctuate desmin ((**F**), arrow) or TIA1 ((**G**), arrow) positivity. (**H**) A fiber with dystrophin aggregates. (**I**) Lack of fiber type grouping pointing away from coexisting reinnervation (ATPase pH 4.3, type 1 fibers stain brown). (This patient was previously reported: Nicolau et al. [24]). Magnification: 40× (**A**–**H**); 10× (**I**).

**Figure 3 ijms-26-02905-f003:**
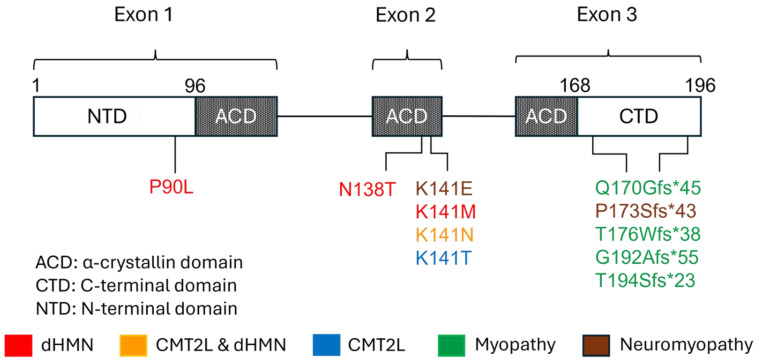
Schematic representation of *HSPB8*, protein domains, and mutations causing neuromuscular diseases. p.P90L, p.N138T, and p.K141M have been associated with dHMN; p.K141T has been associated with CMT2L; p.K141N has been associated with dHMN and CMT2L; p.K141E has been reported in neuromyopathy. p.Q170Gfs*45, p.P173Sfs*43, p.T176Wfs*38, p.T194Sfs*23, and p.G192Afs*55 in CTD have been associated with myopathy; p.P173Sfs*43 has been associated with neuromyopathy.

**Figure 4 ijms-26-02905-f004:**
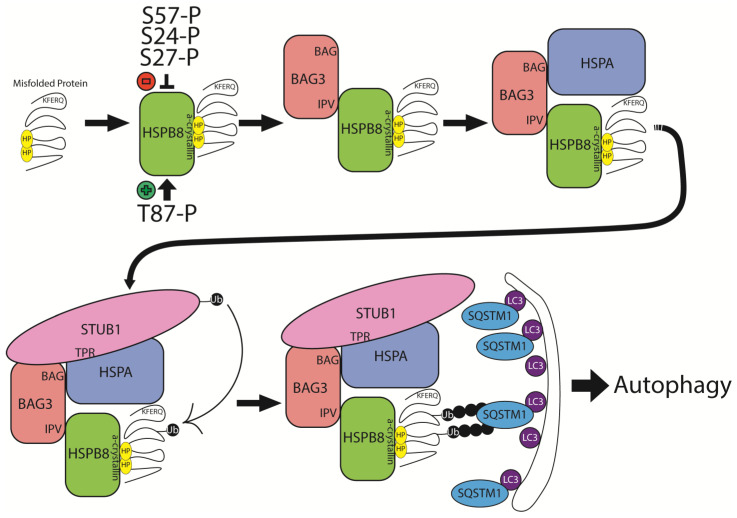
Schematic representation of HSPB8’s structural and functional role in chaperone-assisted selective autophagy (CASA). Hydrophobic (HP) residues are exposed on a misfolded protein and recognized by the alpha-crystallin domain of HSPB8, which is modulated by phosphorylation of key residues T87, S27, S24, and S57. HSPB8 recruits BAG3 through interaction with its IV-X-IV (IPV) motif. HSPA (HSP70) binds the BAG domain of BAG3 and recognizes the KFERQ motif on the misfolded substrate, allowing for further stabilization of the misfolded protein. STUB1 is recruited to the complex, interacting with HSPA through its tetratricopeptide repeat (TPR) domain, and polyubiquitinates the misfolded protein. SQSTM1 recognizes the ubiquitinated substrate and serves as an adaptor protein linking the misfolded protein to the autophagic membrane through interaction with LC3. The misfolded protein subsequently undergoes autophagic degradation.

**Table 1 ijms-26-02905-t001:** Clinical features of *HSPB8*-associated neuromuscular disorders.

Characteristics	dHMN	CMT2L	Myopathy
Age of onset	Childhood to adulthood	Adolescence to adulthood	Childhood to adulthood
Weakness	Distal predominant	Distal predominant	Distal, proximal, or axial predominant
Tendon reflexes	Diminished or absent	Diminished or absent	Normal, diminished, or absent
Muscle atrophy	Present	Present	Present
Cramps	Can be present	Absent	Can be present
Fasciculations	Can be present	Absent	Can be present in neuromyopathy
Skeletal abnormalities	Pes cavus	Pes cavus and scoliosis	Pes cavus, scoliosis, lumbar lordosis

dHMN: distal hereditary motor neuropathy; CMT2L: Charcot–Marie–Tooth type 2L.

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
