# Peer review of "The Spectrum of Small Heat Shock Protein B8 (HSPB8)-Associated Neuromuscular Disorders"

_ijms, 2025, doi:10.3390/ijms26072905_

Round 1

Reviewer 1 Report

Comments and Suggestions for Authors

The authors describe the spectrum of a sHSP (HSPB8) associated with neuromuscular diseases. Since molecular chaperones frequently play essential roles in cellular homeostasis (e.g., proteostasis), their mutants often produce serious situations such as cell death and developmental failure before the onset of the disease. However, the mutation of HSPB8 is associated with dHMN, CMT, and a kind of myopathy. It is an important review because we can know the importance of sHSP. The main issues are shown below.

In the abstract and Introduction, the definition and the term “HSPB8 is a molecular chaperone” should be clarified (“chaperone” is slang). Although the brief explanation about ‘identifying misfolded proteins” is described, it is important to clarify to which category this protein belongs in moelcualr chaperones and what kind of general functions it has.

In the abstract, although they explain that HspB8 is an abbreviation for “Human heat shock protein B8”, this is incorrect. In general, HSP is an abbreviation for heat shock protein. Therefore, sHSP is an abbreviation for the small HSP. In addition, mice have the same gene. Therefore, in this case, “Heat shock protein B8 (HSPB8) is a ubiquitous protein… in various organisms (mammalians), including humans.” may be better.

Whether it is HspB8 or HSPB8, it should be unified in the manuscript.

In Section 4, the general functions of sHSP are not described. These should be included. Moreover, there are some examples of differences from other HSPs, but they are not clear. This could lead to the misunderstanding that all chaperones are involved in these diseases. For example, mutations in HSP70/HSPA are too serious to affect the disease (HSP40/DnaJ is also related to diseases). Only CASA-related molecular chaperones and cochaperones are explained, but it would be better to briefly explain what sHSP is among molecular chaperones and how HSPB8 is classified.

Is there any evidence that when HSPB8 is mutated, it directly affects CASA without taking into account the macro- or other autophagy pathway? This point seems to be a bit of an exaggeration. I think it would be better to revise the sentence.

It would be preferable to add a brief explanation based on the current evidence as to whether or not other small HSPs are related to these diseases.

How about preparing a table that shows at a glance the relationship between the characteristics of the three diseases covered in this review and the HSPB8 mutations?

Author Response

The authors describe the spectrum of a sHSP (HSPB8) associated with neuromuscular diseases. Since molecular chaperones frequently play essential roles in cellular homeostasis (e.g., proteostasis), their mutants often produce serious situations such as cell death and developmental failure before the onset of the disease. However, the mutation of HSPB8 is associated with dHMN, CMT, and a kind of myopathy. It is an important review because we can know the importance of sHSP. The main issues are shown below.

COMMENT 1: In the abstract and Introduction, the definition and the term “HSPB8 is a molecular chaperone” should be clarified (“chaperone” is slang). Although the brief explanation about ‘identifying misfolded proteins” is described, it is important to clarify to which category this protein belongs in moelcualr chaperones and what kind of general functions it has.

RESPONSE 1: We have clarified in the Abstract and Introduction the chaperone category HSPB8 belongs to and provided more details on the protein function especially in the Introduction, and, as requested below, also in Section 4 “Gene and protein structure and function”.

COMMENT 2: In the abstract, although they explain that HspB8 is an abbreviation for “Human heat shock protein B8”, this is incorrect. In general, HSP is an abbreviation for heat shock protein. Therefore, sHSP is an abbreviation for the small HSP. In addition, mice have the same gene. Therefore, in this case, “Heat shock protein B8 (HSPB8) is a ubiquitous protein… in various organisms (mammalians), including humans.” may be better.

RESPONSE 2: The first sentence in the Abstract and Introduction was changed to reflect the Reviewer’s suggestion.

COMMENT 3: Whether it is HspB8 or HSPB8, it should be unified in the manuscript.

RESPONSE 3: We apologize for overlooking the inconsistent terminology. We have corrected the term to “HSPB8” across the manuscript.

COMMENT 4: In Section 4, the general functions of sHSP are not described. These should be included. Moreover, there are some examples of differences from other HSPs, but they are not clear. This could lead to the misunderstanding that all chaperones are involved in these diseases. For example, mutations in HSP70/HSPA are too serious to affect the disease (HSP40/DnaJ is also related to diseases). Only CASA-related molecular chaperones and cochaperones are explained, but it would be better to briefly explain what sHSP is among molecular chaperones and how HSPB8 is classified.

RESPONSE 4: Additional functions of sHSP have been added in Section 4, as well as in Introduction, as recommended above. As this review focuses on neuromuscular disorders caused by HSPB8 mutations, it would be out of scope to review functions of all HSPs and diseases associated with HSP and other sHSP. We however agree with the Reviewer’s observation, and, for further clarification, we have added a small paragraph at the end of Section 4 to indicate the diseases associated with other CASA components, after the existing statement about the crucial role of CASA in the muscle Z-disc maintenance. We have also mentioned other neuromuscular diseases associated with other sHSP.

COMMENT 5: Is there any evidence that when HSPB8 is mutated, it directly affects CASA without taking into account the macro- or other autophagy pathway? This point seems to be a bit of an exaggeration. I think it would be better to revise the sentence.

RESPONSE 5: We believe the Reviewer refers to the statement in the Conclusion. We have modified that sentence to address the Reviewer’s comments.

COMMENT 6: It would be preferable to add a brief explanation based on the current evidence as to whether or not other small HSPs are related to these diseases.

RESPONSE 6: We have added a small paragraph at page 9, at the end of Section 4, regarding the phenotype caused by mutations in other small HSP.

COMMENT 7: How about preparing a table that shows at a glance the relationship between the characteristics of the three diseases covered in this review and the HSPB8 mutations?

RESPONSE 7: The table has been added.

Reviewer 2 Report

Comments and Suggestions for Authors

This manuscript provides a comprehensive overview of the spectrum of neuromuscular disorders associated with HSPB8 mutations. It highlights the role of HSPB8 in protein quality control and discusses the pathomechanisms of different HSPB8-related diseases. The use of animal models to validate the toxic gain-of-function mechanism is innovative.  There are a few more concerns here:

  1. Provide more context on the prevalence and incidence of HSPB8-related disorders to help readers understand the significance of the research.
  2. Include a discussion of potential limitations of the current studies, such as the small sample sizes in some cases or the limitations of animal models.
  3. Discuss how the findings might inform future research directions, such as the development of targeted therapies or the exploration of other genetic modifiers.
  4. Consider adding a section on the potential implications of HSPB8 mutations in other diseases or conditions, as this could broaden the relevance of the paper.

Author Response

This manuscript provides a comprehensive overview of the spectrum of neuromuscular disorders associated with HSPB8 mutations. It highlights the role of HSPB8 in protein quality control and discusses the pathomechanisms of different HSPB8-related diseases. The use of animal models to validate the toxic gain-of-function mechanism is innovative.  There are a few more concerns here:

COMMENT 1: Provide more context on the prevalence and incidence of HSPB8-related disorders to help readers understand the significance of the research.

RESPONSE 2: HSPB8-related neuromuscular disorders are rare. The prevalence of HSPB8-dHMN was already indicated at page 4 (at the end of “Clinical features”). Regarding CMT2L there are no specific numbers in the literature, as its prevalence is often combined with prevalence of HSPB8-dHMN. We have added a sentence about this at page 5, end of section “Clinical features”. The prevalence of HSPB8-myopathy is unknown as only very few families and sporadic cases have been described. In this regard, a sentence was added at page 6, end of section “Clinical features”.   

COMMENT 2: Include a discussion of potential limitations of the current studies, such as the small sample sizes in some cases or the limitations of animal models.

RESPONSE 2: A Section titled “Potential limitations of current studies” has been added just before ‘Conclusion” to address the Reviewer’s comments.

COMMENT 3: Discuss how the findings might inform future research directions, such as the development of targeted therapies or the exploration of other genetic modifiers.

RESPONSE 3: This is now discussed in the newly added Section “Potential limitations of current studies”, inserted before the Section “Conclusions”.

COMMENT 4: Consider adding a section on the potential implications of HSPB8 mutations in other diseases or conditions, as this could broaden the relevance of the paper.

RESPONSE 4: At this point, we can only speculate on potential additional impact of HSPB8 mutations. In this regard, we have elected to add a sentence in the newly created Section “Potential limitations of current studies”, inserted before the Section “Conclusions”.

Round 2

Reviewer 1 Report

Comments and Suggestions for Authors

I have agreed the authors' revision.